# Extrinsic Regulators of mRNA Translation in Developing Brain: Story of WNTs

**DOI:** 10.3390/cells10020253

**Published:** 2021-01-28

**Authors:** Yongkyu Park, Midori Lofton, Diana Li, Mladen-Roko Rasin

**Affiliations:** Department of Neuroscience and Cell Biology, Robert Wood Johnson Medical School, Rutgers University, Piscataway, NJ 08854, USA; mgl74@rwjms.rutgers.edu (M.L.); diana.li1@rutgers.edu (D.L.)

**Keywords:** neocortical development, ligand Wnt3 signaling, Fzd7 receptor, Foxp transcription factor, mRNA translation regulation, morphogen factor

## Abstract

Extrinsic molecules such as morphogens can regulate timed mRNA translation events in developing neurons. In particular, Wingless-type MMTV integration site family, member 3 (Wnt3), was shown to regulate the translation of *Foxp2* mRNA encoding a Forkhead transcription factor P2 in the neocortex. However, the Wnt receptor that possibly mediates these translation events remains unknown. Here, we report Frizzled member 7 (Fzd7) as the Wnt3 receptor that lays downstream in Wnt3-regulated mRNA translation. Fzd7 proteins co-localize with Wnt3 ligands in developing neocortices. In addition, the Fzd7 proteins overlap in layer-specific neuronal subpopulations expressing different transcription factors, Foxp1 and Foxp2. When *Fzd7* was silenced, we found decreased Foxp2 protein expression and increased Foxp1 protein expression, respectively. The *Fzd7* silencing also disrupted the migration of neocortical glutamatergic neurons. In contrast, *Fzd7* overexpression reversed the pattern of migratory defects and Foxp protein expression that we found in the *Fzd7* silencing. We further discovered that Fzd7 is required for Wnt3-induced *Foxp2* mRNA translation. Surprisingly, we also determined that the *Fzd7* suppression of *Foxp1* protein expression is not Wnt3 dependent. In conclusion, it is exhibited that the interaction between Wnt3 and Fzd7 regulates neuronal identity and the Fzd7 receptor functions as a downstream factor in ligand Wnt3 signaling for mRNA translation. In particular, the Wnt3-Fzd7 signaling axis determines the deep layer Foxp2-expressing neurons of developing neocortices. Our findings also suggest that Fzd7 controls the balance of the expression for Foxp transcription factors in developing neocortical neurons. These discoveries are presented in our manuscript within a larger framework of this review on the role of extrinsic factors in regulating mRNA translation.

## 1. Introduction

### 1.1. Neocortical Development

In evolutionarily advanced species, the neocortex develops into the central neuronal circuit of higher cognitive function and voluntary motor behavior [1,2,3,4]. The development of this complex laminar structure requires an intricate progression of spatially and temporally controlled molecular events starting from neural stem cells [2,4,5,6,7,8,9,10]. The disruption of these events can lead to abnormal neocortical development associated with neurological and psychiatric disorders, such as epileptic encephalopathy (EE), autism spectrum disorder (ASD), and intellectual disability (ID), which are often expressed simultaneously [11,12]. The neocortical structure gives rise to six horizontal layers (I to VI), segregated principally by cell type and the specific site of neuronal connections. The neocortex, like all other central nervous structures, contains two main types of cells, neurons and glia, along with endothelial cells and microglia. Within the neuronal subtypes, the highly organized positioning of distinct subtypes of glutamatergic projection neurons (also known as pyramidal neurons) defines each layer during neocortical development. Each layer’s pyramidal neurons form specific axonal projections, with different targets determining the canonical circuitry within the cortex [1,2,7,13,14]. Pyramidal neurons in the upper layers II and III solely project their axons to other areas within the neocortex (intracortically within the ipsilateral hemisphere or to contralateral targets via the corpus callosum). In contrast, pyramidal neurons in deeper layers V and VI predominantly project out of the cortex to sites such as the thalamus, brainstem, and spinal cord. Interestingly, glutamatergic neurons in layer IV make short-range, local connections to other cortical layers. In addition to the differentiation in their efferent projections, each layer has distinction in their dendritic structures, incoming afferent axons, and transcription factor identities [4]. For example, upper layer glutamatergic neurons express transcription factors Satb2, Brn1, Brn2, Cdp, Cux2, Svet1, and Foxp1 while lower layer neurons express Fezf2, Tbr1, Otx1, Ctip2, and Foxp2 [4].

The primary stem cell source of neocortical projection neurons is radial glial progenitor cells (RGCs), widely known as scaffolds that guide the migration of new neurons from their birth zones close to the ventricles, towards their final destination of the cortical plate [15]. RGCs are multipotent, as they possess the ability to produce diverse types of glutamatergic neurons and glia as well as other progenitor subtypes; they can also proliferate extensively by self-renewal. These properties mark RGCs as a special type of neural stem cell (NSC) in the developing neocortex [3,16], of which the RGCs reside in the ventricular zone (VZ) adjacent to the lateral ventricles (LV). RGCs can give rise to a post-mitotic neuron either directly or indirectly via an intermediate progenitor (IP) cell. IPs populate the space basal to the VZ, known as the subventricular zone (SVZ). Embryonic cortical neurons are generated in an inside-out fashion to populate the six layers of the cortex [2,4,5,6,7,8,9,10]. Earlier-born neurons populate the deepest of the six cortical layers (V–VI), while later born neurons populate progressively more superficial layers (II–IV). These layers contain distinct neuronal subtypes that differ based on morphology, electrophysiological activity, axonal connectivity, and gene expression [17].

### 1.2. Post-Transcriptional Regulation in Neocortical Development

To achieve a six-layered laminar neocortex, it must go through genesis, migration, and maturation, regulated by transcriptional, post-transcriptional and epigenetic mechanisms. [2,4,5,6,7,8,9,10,18,19,20,21]. Recent discoveries of transcriptional regulation combined with advances in nucleic acid-predicated technologies have spearheaded a series of studies analyzing the neocortex’s genome and transcriptome across its neocortical layers [22,23]. Investigations on neurological disorders involving the neocortex, such as autism spectrum disorders, highlight the transcriptional framework and concentrate on abnormalities at the genomic and transcriptomic levels [24,25,26,27,28,29,30,31,32]. The precise equilibrium between RGC self-renewal and neuronal differentiation in the developing neocortex is modulated by the intricate network of intrinsic cellular programs and mechanisms. Experiments modulating gain- and loss-of-function show that transcription factors Sox5, Satb2, Fezf2, Tbr1, and Ctip2 comprise a cross-repressive transcriptional circuit that regulates projection neuron specification [17,33,34]. Subsequent studies have identified many additional subtype-specific genes, including transcription factors Brn1/2 and Rorb, which play critical roles in the specification and establishment of the neocortical layers [22,35,36]. Additionally, other studies have recognized the defining epigenetic signatures across differentiating cortical areas during development, evolution, aging, and disease [37,38,39,40,41]. Recent works have shown the importance of de novo somatic mutations, which occur post-zygotically and, thus, are present only in a subset of cells of an affected individual [42]. These somatic mutations contribute to neurodevelopmental diseases, including epileptic encephalopathies, intellectual disability, and autism spectrum disorder. Collectively, these studies support the notion that transcriptional regulation is a critical determinant of mature neuronal subtypes.

Despite the fact that transcription is considered the primary predictor of protein levels under steady state conditions, recent evidence shows that post-transcriptional control plays a dominant role during short-term state transitions, such as differentiation in response to extrinsic stimuli [43,44,45,46]. In order to better comprehend the molecular basis of the vast complexity and diversity encompassing neocortical development, post-transcriptional mechanisms have been characterized, including RNA-binding proteins (RBPs), ribosomal proteins (RPs), micro-RNAs (miRNAs), and long non-coding RNAs (lncRNAs). Understanding the regulation at the post-transcriptional level brings clarity to the transcriptional complexity and spatiotemporal specification of neuronal subtypes generated from RGCs in the developing neocortex [2,4,5,6,7,8,9,10,18]. RBPs play a crucial role in driving neocortical development by controlling the steps of post-transcriptional processing (splicing, transport, stability/decay, localization, and translation) [4]. RBPs and miRNAs regulate groups of functionally related mRNAs along a coordinated pathway of RNA processing, allowing cells to respond with unprecedented efficiency to extrinsic signals [47]. The combination of ribosomal proteins associated with translating polysomes is dynamic in both time and space during cortical development [48]. While the levels of certain mRNAs encoding transcriptional and translational regulators remain constant throughout cortical development, the differentially associated translating polysomes are temporally specific in manner, suggesting that the posttranscriptional regulators themselves could also be subject to regulation [48].

*Cis*- and *trans*-encoded translation regulations below significantly influence the fate of a cell. (1). In response to environmental cues, ribosomes are positioned locally in radial glial endfeet to translate mRNAs. (2). Upstream open reading frames (uORFs) repress canonical ORF translation by causing decay, leading to reduced translational efficiency. (3). MicroRNAs incorporated into an Argonaute complex bind to the 3′UTR to repress target mRNA expression. (4). RBPs and RNA helicases either promote or repress mRNA translation. (5). mRNAs and proteins are segregated in dividing cells, driving their asymmetrical inheritance in newborn cells. (6). RNA modifications, such as m6A, may affect mRNA stability and mRNA translation. (7). Microexons can change the reading frame or encode premature stop codons to modulate protein expression. These mechanisms are present and active in progenitors and neurons (reviewed in [9]). It is noted that RBPs play critical roles in the translation of NSC cell fate decisions. The RBP human antigen R (HuR) controls the association between translating polysomes and functionally-related mRNAs encoding transcriptional, translational, and neuronal layer-specific regulators in a developmental stage-specific manner [18,49]. HuR also controls the composition of translational initiation factors, elongation factors, and combinatorial patterns of ribosomal proteins within polysomes [49]. HuR interacts with both common and unique subsets of mRNAs during early and late neurogenesis, suggesting that mRNA targets of RBPs evolve as a function of time [50]. Recent interesting work demonstrates that upstream RBP, Celf1, regulates the 5′ UTR-driven isoform-specific translation of another RBP, Elavl4, which results in the specific development of glutamatergic neurons [12]. These findings reveal a dynamic interplay between distinct RBPs and alternative 5′ UTRs in neuronal development. RBP Fragile X Mental Retardation Protein (FMRP) is another translational regulator that plays a significant role during corticogenesis. At RGC basal endfeet, FMRP integrates mRNAs encoding signals and cytoskeletal factors which are associated with autism and neurogenesis [51]. FMRP represses the RGC to IP transition, maintaining RGCs in an undifferentiated state [52]. Thus, these reports reveal how RBPs are important in regulating multiple aspects of embryonic cortical development including mRNA splicing, stability, localization, and translation.

Multipotent RGCs are transcriptionally primed, allowing the rapid generation of diverse progeny. Post-transcriptional regulation, including translation, refines gene expression to specify cell fates [9]. RGCs express mRNAs that maintain stem cell proliferation and self-renewal in the stages of cortical embryonic development. In addition, they also express mRNAs that help to promote neurogenesis (transcriptional priming or pre-patterning) [53]. During the stem cell state, translational repression of mRNAs interferes with neuronal cell differentiation: elF4E and its binding partner 4E-T compose a repressive complex that sequesters and represses proneurogenic mRNAs expressed in RGCs [54]. The 4E-T protein targets mRNAs via RBPs, such as Smaug2, in an undifferentiated state [55]. In order to generate neurons, the disruption of these relevant translational repression complexes occurs with external proneurogenic signals by mechanisms that remain unknown, enabling the translation of mRNAs mediating neurogenesis. Nanos1, an RBP that promotes neurogenesis presumably by repressing self-renewal genes, suggests that appropriate neuronal differentiation requires both the translational derepression of proneurogenic mRNAs as well as the *de novo* repression of mRNAs associated in stem cell maintenance [55].

### 1.3. Neocortical Development Orchestrated with Extracellular Signals

RGCs possess an elongated radial morphology that allows it to access extrinsic signals originating from the meninges, vasculature, newborn neurons, cerebrospinal fluid, and ingrowing axons; many of which regulate RGC cell fate decisions [56,57]. Collective proteomic analyses from the cell-surface of RGCs and newborn neurons have revealed rich growth factors in the developing cortex, including many previously uncharacterized autocrine and paracrine interactions [57]. Extrinsic signals generically control RGC proliferation and differentiation while regulating the specification of particular neuronal subtypes [58]. Namely, epidermal growth factors (EGFs) influence several cells including neural stem cells (NSCs), oligodendrocytes, astrocytes, and neurons with different effects on their proliferation, migration, and differentiation [59]. In multiple neuronal systems, extracellular factors can also regulate mRNA translation specificity [60,61]. For example, nerve growth factor (NGF) stimulates the translation of eukaryotic elongation factor 1A-1 (eEF1A-1) mRNA by specifically recruiting it to polyribosomes in neuronally differentiated PC12 cells [60]. Brain-derived neurotrophic factor (BDNF) also regulates the translation of a select group of mRNAs during neuronal development, especially within neuronal dendrites in the mammalian target of rapamycin (mTOR)-dependent pathway [61].

Timed extracellular signaling events are essential in determining normal neocortical development. One major developmental event is the ingrowth of thalamic axons at mid-neurogenesis [62]. The thalamus, a brain structure in the vertebrate diencephalon, plays a central role in regulating diverse functions of the cerebral cortex with guidance mechanisms for thalamocortical axons (TCAs) [63]. When the mouse thalamus undergoes embryonic neurogenesis, several Wnt ligands, such as WNT3, WNT3A, and WNT7B, are expressed [48,64] and involved in the development and control of TCA projection in thalamic glutamatergic neurons [65]. The TCAs grow through the subpallium and reach the cortex by E14.5 [66]. It was shown that the E14.5 mouse thalamus produces a diffusible factor that promotes the proliferation of cortical precursors over a restricted developmental window [67], suggesting that thalamic afferents control the cortical area size by promoting the division of a specific population of neural progenitor cells. The timed ingrowth of thalamocortical axons is accompanied by the secretion of extracellular factors into the developing neocortex; however, the underlying cellular and molecular mechanisms still remain unknown.

### 1.4. WNT Signaling in Neuronal Diseases

The WNT signaling pathway is an evolutionarily conserved signal transduction pathway that regulates a wide range of cellular functions including cell proliferation, cell fate determination, apoptosis, cell migration, and cell polarity during development and stem cell maintenance in adults [68,69,70]. WNT proteins are lipid-modified glycoproteins that are about 350–400 amino acids in length [71] and act as ligands that interact with Frizzled (FZD) receptors which are located on the cell surface, to activate intracellular signaling pathways [71,72,73]. FZD receptors are a family of G protein-coupled receptor proteins that have seven-pass transmembrane domains and act as the primary receptors for WNT signaling. Once FZD receptors are activated, a signal is intracellularly transduced causing the activation of protein Disheveled (Dvl or Dsh), which induces the WNT signal branching off into multiple downstream pathways that can be categorized into the canonical WNT/β-catenin pathway and the non-canonical WNT pathway [73,74,75]. During canonical WNT signaling, β-catenin is released from scaffolding proteins, such as Axin, and helps accelerate messaging between the cell membrane and nucleus [76]. β-catenin is then localized to the nucleus where it forms a complex with DNA bound TCF/LEF transcription factor resulting in the activation of WNT-responsive genes [74,77,78,79]. Non-canonical pathways are independent of β-catenin signaling and possess the ability to activate additional signals that regulate several cellular behaviors, including planar cell polarity, cell movements during gastrulation, and the cell migration of neural crest cells [80,81,82,83].

In addition to embryonic development and cancer progression, the WNT pathway has been identified as a key contributor to Alzheimer’s (AD) and metabolic diseases. The WNT pathway is strongly linked to AD pathogenesis because the pathway is fundamental to the development of the central nervous system [84]. Several studies have reported a causative link between WNT signaling and autism. Mice mutants for *Dvl1* and *Dvl3* showed reduced β-catenin expression causing premature deep layer neurogenesis of neural progenitors in precise brain regions during embryonic development. This expressed an adverse effect on the establishment of neural connections in the future prefrontal cortex, which was observed with serious deficits in brain size and social behavior in adults [85]. The administration of a GSK-3 inhibitor to activate the canonical WNT pathway *in utero* can reverse these deficits caused by diminished β-catenin expression. WNT signaling is also linked to the pathogenesis of Parkinson’s disease via the dysregulation of the expression of WNT pathway components in brain tissue of Parkinson’s disease patients [86]. Further studies revealed that activation of the WNT/β-catenin pathway in astrocytes could recover neighboring midbrain dopaminergic neurons after post-injury in vivo. Thus, it can serve as a therapeutic approach to promote recovery of midbrain dopaminergic neurons [87,88]. Patients with schizophrenia show altered GSK3 activity [89] as well as increased expression of WNT1, which can lead to synaptic rearrangement and plasticity [90]. Additionally, several SNPs in FZD3 were reported to be associated with susceptibility to schizophrenia [91]. The relevance of WNT signaling in depression, bipolar disorder, epilepsy, seizures has also been reported [92].

### 1.5. mRNA Translations Regulated by WNTs Signaling

Neural stem cell (NSC) niches are abundant in extrinsic signals that regulate their proliferation, differentiation, and cell fate specification. These environmentally driven signaling pathways make NSC likely to modulate RBP expression and/or its interactions with target mRNAs [57]. WNT signaling is related to Fragile-X syndrome (FXS), a heritable form of autism caused by a deficiency of RBP Fragile-X mental retardation protein (FMRP), resulting in variable degrees of autism-like behavior. FMRP functions as a negative regulator of protein translation at the RGC basal endfeet. FXS patients showed reduced expression of WNT7a and reduced β-catenin-dependent signaling, and in the brain of *Fmr1* (FMRP gene) knockout mice the translation of *Wnt2* mRNA was found to be reduced [93,94,95,96]. The connection of FMRP to WNT signaling was extended with the discovery of β-catenin and FMRP complex functioning in WNT-dependent translation regulation [97]. The interaction between β-catenin and FMRP leads to a basal recruitment of β-catenin to the messenger ribonucleoprotein and translational pre-initiation complex, fulfilling a translational repressor function (Figure 1). Wnt3a stimulation provokes this function in part by separating β-catenin from the pre-initiation complex (Figure 1) [97]. Another example of how extracellular WNT regulates cytoplasmic mRNA translation is shown in Wnt1 signaling, which enhances embryonic NSC maintenance and proliferation by promoting RBP Imp1 expression during cortical development (Figure 1) [98]. Not only does it silence proneurogenic mRNAs, the Imp1 also translationally increases the stability and expression of pro-self-renewal mRNAs, such as *Hmga2*, in order to maintain NSCs in an undifferentiated state (Figure 1) [98]. Consistent with these functions of Imp1 from Wnt1 signaling, the cortex specific Imp1 ablation resulted in reduced NSC self-renewal and accelerated neuronal and glial differentiation [98].

Transcription factors, such as Foxp2, that promote layer-specific neuronal differentiation are highly regulated during development [99]. In our previous study, it was found that timed Wnt3 secretion from thalamocortical axons around mid-neurogenesis regulates the composition of ribosomal proteins, including Rpl7, in the subset of mRNAs that are associated with polysomes (Figure 1) [48]. Wnt3 promotes *Foxp2* mRNA association and translation in polysomes through the regulation of its 3′UTR (Figure 1). The conditional ablation of thalamic Wnt3 causes damage to the specification of Foxp2-postive deep layer neurons [48]. How do these extrinsic cues promote and support post-transcriptional machinery? Many RBPs contain multiple consensus phosphorylation sites, implying that they can be regulated by phosphorylation [100]. For instance, HuR phosphorylation is important for regulating the translational control of the autism-associated mRNAs, *Foxp1* and *Foxp2*, in the developing cortex [50]. For accurate responses to growth factor stimuli in several other environments, the receptor-mediated phosphorylation of RBPs might provide a mechanism that allows the rapid and reversible alteration of the translation subsets of mRNAs which regulate NSC biology.

However, the WNT receptor that may mediate these mRNA translation events remains unknown. In neurogenesis of neocortical cells, it is critical to discover which receptor transfers the extrinsic Wnt signaling arriving from the thalamus into the neocortex, to regulate the mRNA translation in developing neocortical neurons. Here, we demonstrate that Frizzled member 7 (Fzd7) as a Wnt3 receptor functions downstream in the Wnt3-regulated translation. The Fzd7 co-localizes with Wnt3 in developing neocortices and is co-expressed in neurons expressing Foxp1 or Foxp2 in the distinct layers of neocortices. The silencing and overexpression of *Fzd7* resulted in changes in the Foxp1 and Foxp2 protein expressions, respectively. We further found that Fzd7 is required for the Wnt3 induced *Foxp2* mRNA translation via 3′UTR, while the novel effect on *Foxp1* is not Wnt3 dependent. In conclusion, we show that the interaction between Wnt3 and Fzd7 regulates neuronal identity and that Fzd7 tightly controls the expression of Foxp transcription factors in developing neocortical neurons.

## 2. Methods

### 2.1. Animals and In Utero Electroporation

For all embryonic experiments, mice of both genders were used without sex determination. Experiments involving animals were carried out in accordance with Rutgers University Medical School’s Institutional Animal Care and Use Committee. For timed pregnancies, adult pregnant female CD-1 mice were purchased from Charles River Laboratories. In utero electroporation (IUE) was performed at E13 (*Fzd7* shRNA) or E16 (*Fzd7* OE) and analyzed at P0 as previously described [101]. Co-electroporation was accomplished with 1 µL of mixed plasmids containing 4 µg/µL of control or *Fzd7* shRNA/OE vector along with 1 µg/µL of CAG-GFP reporter (4:1 ratio). For each IUE, at least three transfected neocortices were used in experiments and this is indicated in figure legends with the “*n*” value. Three to five images from different sections of each transfected neocortex were used for the quantifications of immunostaining.

### 2.2. Primary Neuronal Culture and N2a Cell Transfection/Culture

Alternating E13 neocortices in the same litter were co-electroporated (IUE) with *Ctrl* shRNA and CAG-GFP or *Fzd7* shRNA and CAG-RFP plasmids. Four hours post-IUE, the primary neuronal cells were mixed and in vitro cultured together [12]. The primary neuronal cultures were performed as described [48]. Neuroblastoma N2a cells (ATCC #CCL-131, Manassas, VA, USA) were grown in Dulbecco’s modified eagle medium (Gibco #31053-036, Waltham, MA, USA) containing 10% FBS (Gemini #900-108, West Sacramento, CA, USA), 1% GlutaMAX (Gibco #35050-061, Waltham, MA, USA), 1% Sodium Pyruvate (100mM, Gibco #11360-070, Waltham, MA, USA), and 1% Penicillin/Streptomycin (Corning #30-002-CI, Corning, NY, USA). TrypLE Express (Gibco #12604-021, Waltham, MA, USA) dissociation reagent was used for regular maintenance. Lipofectamine 2000 (Invitrogen #11668-019, Waltham, MA, USA) was used to perform transfections as per the manufacturer’s protocol.

### 2.3. Plasmids

The shRNA and overexpression plasmids were commercially obtained from OriGene; shRNA: control (#TR30013, Rockville, MD, USA) and *Fzd7* (#TG500742B, Rockville, MD, USA); overexpression: control (pCMV6-AC, #PS100020, Rockville, MD, USA) and Fzd7 (NM_008057, #MC202423, Rockville, MD, USA). For the *Luciferase*-*Foxp2*_*3′UTR* construct, the full-length of *Foxp2_3′UTR* (3922 bp) was subcloned to the downstream of a firefly *Luciferase* cassette in the pcDNA3.1-*Luciferase* plasmid as previously described [48].

### 2.4. Luciferase Reporter Assay

N2a cells were seeded on 12-well plate and co-transfected the next day with 0.4 µg of the *Luciferase*-*Foxp2*_*3′UTR* reporter vector and 2 µg of the *Ctrl* shRNA or *Fzd7* shRNA plasmids per one well. Twenty-four hours after transfection, cells were split into three experimental groups: (1) mock treated (base media only), (2) treated with 100 ng/mL of recombinant Wnt3 (Abnova #H00007473-P01, Taipei, Taiwan), and (3) treated with 100 ng/mL of both Wnt3 and SFRP1 (R&D Systems #5396-SF-025, Minneapolis, MN, USA) for 48 h. After washing with PBS, cells were lysed at room temperature for 15 min with 250 µl of 1X Passive Lysis Buffer (Promega # E1910, Madison, WI, USA), then divided equally into two fractions. One fraction was followed by the Pierce™ 660nm Protein Assay (Thermo Scientific # 22660, Waltham, MA, USA) and Luciferase measurement with the Dual-Luciferase Reporter Assay System (Promega #E1910, Madison, WI, USA). The other fraction of cells was used for total RNA isolation using TRIzol LS (Invitrogen # 10296028, Waltham, MA, USA) for qRT-PCR. Relative Luciferase light units (RLUs) were normalized to the *Luciferase* mRNA levels measured in duplicate qRT-PCR run in parallel for each condition.

### 2.5. Quantitative Real-Time PCR

Total RNAs were isolated using TRIzol LS (Invitrogen #10296028, Waltham, MA, USA) following the manufacturer’s instructions, while residual DNAs were removed by incubating the RNAs with the Turbo DNA-free™ kit (Invitrogen #AM1907, Waltham, MA, USA). The Applied Biosystems StepOne real-time system and reagents (TaqMan™ RNA-to-Ct™ 1-step kit, Thermo Fisher #4392938, Waltham, MA, USA) were used to perform qRT-PCR with 50 ng of RNA per reaction and commercially available TaqMan probes of control and target mRNAs below. The results were analyzed using the ∆∆Ct method normalized with control *Gapdh/ActB* mRNAs [102]. TaqMan FAM probes (Thermo Fisher, Waltham, MA, USA): *Gadph*, Mm99999915_g1; *ActB*, Mm01205647_g1; *Wnt3*, Mm00437336_m1; *Fzd7*, Mm00433409_s1; *Luciferase*, Mr03987587_mr.

### 2.6. Primary Antibodies

The following primary antibodies and dilutions were used for immunocytochemistry: Wnt3 (1:250, rabbit; LifeSpan BioSciences #LS-C774904, Seattle, WA, USA), Fzd7 (1:50, goat, Thermo Fisher #PA5-47232, Waltham, MA, USA), Foxp1 (1:1000, rabbit; Abcam #ab16645, Cambridge, MA, USA), Foxp1 (1:250, mouse; lab stock), Foxp2 (1:250, rabbit; Abnova #PAB12684, Taipei, Taiwan), Foxp2 (1:250, goat; Santa Cruz Biotechnology #sc21069, Dallas, TX, USA), Gapdh (1:300, mouse, Millipore Sigma #MAB374, St. Louis, MO, USA), GFP (1:1000, chicken; Aves Labs #GFP-1020, Tigard, OR, USA), RFP (1:1000, rabbit; Antibodies-Online #ABIN129578, Limerick, ME, USA). Appropriate species-specific Donkey secondary antibodies (Jackson ImmunoResearch, West Grove, PA, USA) were used at a 1:250 dilution.

### 2.7. Immunohistochemistry and Confocal Imaging

Immunohistochemistry was performed as previously described [49]. Briefly, embryonic brains from postnatal day 0 (P0) were dissected and fixed by immersion for 8 h with 4% paraformaldehyde (PFA, Sigma-Aldrich #158127, St. Louis, MO, USA) in 1X PBS, pH 7.4. After fixation, the brains were washed three times in 1× PBS and coronally-sectioned at 70–80 µm using a Leica VT1000S vibratome. Free-floating sections were then incubated in blocking solution (5% Normal donkey serum (Jackson ImmunoResearch #017-000-121, West Grove, PA, USA), 1% Bovine serum albumin (Biomatik #A2134, Cambridge, ON, Canada), 0.1% Glycine (BDH #BDH4156, Radnor, PA, USA) and 0.1% Lysine (Sigma #L5501, St. Louis, MO, USA) in 1X PBS, pH 7.4) and 0.4%Triton (Sigma-Aldrich #X100, St. Louis, MO, USA), gently shaking for 2–4 h at room temperature. Afterwards, the sections were immediately transferred to a solution of primary antibody resuspended in blocking solution +0.4%Triton X-100, and incubated while gently rocking overnight at 4 °C. The following day, sections were washed 3 times in 1X PBS and incubated with secondary antibody diluted in blocking solution without Triton X-100, gently shaking at room temperature for 1–2 h. Next, sections were washed again three times in 1X PBS before being incubated in 1 μg/mL of DAPI (Fisher Scientific #D1306, Waltham, MA, USA) for 10 min. Finally, sections were washed two more times in 1× PBS before being mounted with Vectashield mounting media (Vector Laboratories #H1000, Burlingame, CA, USA). The immunostaining of Wnt3 (ligand protein outside of membrane) was performed without Triton X-100. Primary neuronal and N2a cells were cultured on glass coverslips, with which immunocytochemistry was followed as written above. Then, slides were imaged with an Olympus BX61WI confocal microscope using 10×, 20×, or 60× objective lens and processed using Fluoview FV-1000. All images used in analysis were taken with the same confocal settings per experiment to allow for accurate comparisons of fluorescent intensity.

### 2.8. Immunohistochemistry Quantification and Statistical Analysis

Ten equal sized bins were framed in each confocal image, of which markers/cells were quantified by separately blinded investigators using Gimp2 software to calculate the percentage of Wnt3/Fzd7/Gapdh or Foxp1/Foxp2-positive neurons among all GFP/RFP cells. The graph bars represent mean and ± SEM, as noted in the figure legends. The number (*n*) of replicates for each experiment and the appropriate statistical test are noted in figure legends. For example, with three biological replicates (*n* = 3 animals), we made three to five images from different sections of each transfected neocortex. Then, blinded investigators counted the markers/cells from each image independently. It was not known to them which image was derived from which animal. Statistical significance (Student’s *t*-test) is reported as: *: *p* < 0.05, **: *p* < 0.01, ***: *p* < 0.005.

All materials available upon request. Contact Dr. Mladen-Roko Rasin (roko.rasin@rutgers.edu).

## 3. Results

### 3.1. Wnt3 Morphogen and Fzd7 Receptor Are Co-Localized with Foxp1 and Foxp2 Transcription Factors in Developing Neocortices at E16 and P0

To characterize how thalamic morphogen Wnt3 regulates neocortical mRNA translation, we aimed to identify its receptor. It was reported that Wnt3 binds to the Frizzled 7 (Fzd7) receptor in intestinal stem cells [103]. In addition, the functional interaction between Wnt3 and Fzd7 led to the activation of the Wnt/β-catenin signaling pathway in hepatocellular carcinoma cells [104]. It is known that *Fzd7* mRNA is expressed in the developing neocortex. These findings suggested that the action of Wnt3 may be mediated through the Fzd7 receptor in the developing neocortex. Therefore, we first established the expression sites of Fzd7 protein using immunohistochemistry (Figure 2). At E16, the Fzd7 receptors were expressed evenly throughout the cortical plate (CP) (Figure 2A) and colocalized with Wnt3 ligands (arrow in Figure 2A). Furthermore, the Fzd7 protein overlapped in both upper layer and lower layer neurons of the CP expressing Foxp1 and Foxp2, respectively (Figure 2A). The Wnt3/Fzd7 and Fzd7/Foxp1/Foxp2 colocalizations were still present at P0 (Figure 2B). Taken together, these observations suggest that the Wnt3 secreted from the thalamus at E15 [48] can interact with the neocortical Fzd7 receptor in order to timely regulate the protein synthesis of the neocortical Foxp transcription factors.

### 3.2. Wnt3 Interacts with Fzd7 in Neuroblastoma N2a Cells

If the extracellular Wnt3 interacts with the Fzd7 receptor in the cell membrane, *Fzd7* downregulation should reduce Wnt3 cell membrane immunostaining. To test this, neuroblastoma N2a cells were transfected with either Control (*Ctrl*) or *Fzd7* shRNA plasmid containing *Green Fluorescent Protein* (*GFP*) gene (Figure 3). The transfection efficiencies of N2a cells in culture dish with shRNA plasmids (GFP cells) illustrated similarities between *Ctrl* shRNA (36.0%) and *Fzd7* shRNA transfection (34.2%) (Appendix A). However, *Fzd7* mRNA expression was reduced by 68% in total cells with *Fzd7* shRNA transfection (Figure 3B), suggesting efficiency of the *Fzd7* shRNA in hand. After 3 days of each shRNA transfection, Wnt3 was added into the culture medium at 100 ng/mL that was reported to be sufficient amount to promote Foxp2 translation in neocortical projection neurons [48]. The *Fzd7* shRNA transfected GFP cells showed a reduced Fzd7 intensity compared to the *Ctrl* shRNA transfected GFP cells (arrows in Figure 3A). The difference in Fzd7 intensity was also observed between the *Fzd7* shRNA transfected GFP cells (arrows) and non-transfected cells (arrowheads) from the same culture dish (lower panel of Figure 3A). The Fzd7 signal intensity was reduced by 54.2% in the *Fzd7* shRNA transfected GFP cells when it was normalized to the signal from the *Ctrl* shRNA transfected GFP cells (Figure 3C). Thus, these data indicate a successful and significant decrease of Fzd7 protein expression by the shRNA in hand.

We then analyzed the interaction between Wnt3 and Fzd7 at the cell membrane using immunohistochemistry after 4 h of Wnt3 treatment (Figure 3A), with which Wnt3 intensity became clear and strong at the cell membrane compared to no treatment of Wnt3 (Appendix A). Remarkably, the Wnt3 intensity at the cell membrane also decreased in the *Fzd7* shRNA transfected cells when compared to the *Ctrl* (Figure 3A,D). This finding correlated with the decrease of Fzd7 expression (Figure 3B,D). The intensity of Gapdh protein was used as a negative control and remained unchanged between the *Ctrl* and *Fzd7* shRNA transfected GFP cells (Figure 3E and Appendix A). We also confirmed that the *Fzd7* shRNA did not affect the endogenous Wnt3 mRNA expression (data not shown). These data suggest that the downregulation of Fzd7 receptor reduces the interaction of Wnt3 ligand with the cell membrane.

### 3.3. Fzd7 Silencing Results in Changes of Foxp Transcription Factors on Primary Neuronal Cells

To characterize the roles of Fzd7 receptor in the neocortical development with the Wnt3 effect on the translation of Foxp transcription factors, we first in utero electroporated (IUE) *Ctrl* or *Fzd7* shRNA plasmid with either *CAG*-*GFP* or *CAG*-*RFP* reporter, respectively [2,12,50,105]. *Ctrl* shRNA/*CAG-GFP* or *Fzd7* shRNA/*CAG-RFP* were transfected in the neocortices of separate embryos of the same litter at E13 and the primary neuronal cell cultures were made together from all neocortices on the same day [12]. After four days in vitro (DIV), the intensities of Foxp proteins were quantified between GFP^+^ (*Ctrl* shRNA) and Red Fluorescent Protein positive (RFP^+^) (*Fzd7* shRNA) cells. Intensities of Foxp protein expression were normalized to the intensity (+) observed in non-transfected cell as an intrinsic control of immunostaining quality (Figure 4A). The strong (++) intensity of Foxp2 protein in *Fzd7* shRNA cells (arrow: GFP cells; arrowhead: RFP cells in the upper panel of Figure 4A) decreased upon *Fzd7* silencing (Figure 4B). In contrast, the strong (++) intensity of Foxp1 staining (lower panel of Figure 4A) increased upon *Fzd7* silencing (Figure 4C). These data suggest the Fzd7 receptor regulates both the Foxp2 and Foxp1 expressions in developing neocortical neurons, but in opposing directions. Fzd7 presence promotes Foxp2 expression and suppresses Foxp1 expression.

### 3.4. Fzd7 Downregulation Disrupts Neuronal Migration and Regulates Foxp2 and Foxp1 Protein Expression In Vivo in Contrasting Ways

Next, we investigated whether the *Fzd7* downregulation in vivo affects the Foxp2 and Foxp1 protein expressions in developing neocortical neurons in a similar fashion as we found in vitro. Either *Ctrl* or *Fzd7* shRNA plasmid was electroporated in utero with a *CAG*-*GFP* reporter at E13 and the GFP^+^ cells in transfected neocortices were analyzed at P0 [12]. When *Fzd7* was silenced, the GFP+ axons projected appropriately for the developmental stage and were comparable to the *Ctrl* (data not shown). However, the distribution of GFP+ cells across 10 bins in the neocortical wall was different between the *Ctrl* and *Fzd7* shRNA (Figure 5A,B). In particular, we found significantly more GFP^+^ cells in the upper bins (Bin 1–5) than in the lower bins (Bin 6–10) when *Fzd7* was silenced (Figure 5A,B).

Consistent with the in vitro data (Figure 4A,B), the in vivo Foxp2 expression was also reduced when *Fzd7* was downregulated (Figure 5C,D). The change in the Foxp2 expression affected by *Fzd7* silencing was most pronounced in bins 4–5 (arrow in Figure 5C and Appendix A). In contrast, the Foxp1 expression in vivo increased when *Fzd7* was silenced (Figure 5C,E), similar to our in vitro findings (Figure 4A,C). Notably, the increase in Foxp1^+^/GFP^+^ cells was observed larger in the lower bins 6–10 (Figure 5E), even though Foxp1 is predominantly expressed in the upper layers of the neocortex (Figure 2B and Figure 5C). In particular, the highest increase in Foxp1 expression was seen in bins 6–7 (arrowhead in Figure 5C and Appendix A). It is plausible that the disrupted migration of GFP^+^ cells that were *Fzd7* silenced (Figure 5A,B) influences the numbers of Foxp2^+^ or Foxp1^+^/GFP^+^ cells (Figure 5C–E). However, the bins showing a bigger change in either Foxp2 or Foxp1 expression (arrow or arrowhead in Figure 5C and Appendix A) are located to where Fzd7 was found to be normally expressed (Figure 2B). Therefore, taken together with the in vitro findings, the data above further suggest that the Foxp2 and Foxp1 expressions in developing neurons of these regions are regulated by Fzd7 signaling.

### 3.5. Fzd7 Overexpression Altered Foxp2 and Foxp1 Protein Expressions in a Layer-Specific Fashion

Since *Fzd7* silencing increased the Foxp1 expression present in upper layer neurons, we subsequently hypothesized that *Fzd7* overexpression in upper layers will decrease Foxp1 expression. To test this, we in utero electroporated *GFP* reporter and either *Ctrl* or *Fzd7* overexpression (OE) plasmid at E16 to selectively target upper layer neurons [10], and these transfected neocortices were analyzed at P0 [12]. When *Fzd7* was overexpressed, GFP^+^ cells were reduced in upper bins 1–5 and increased in the lower bins 6–10 (Appendix A). This abnormal distribution of GFP^+^ cells (Appendix A) represented that *Fzd7* overexpression affects the migration of neuronal GFP cells (Appendix A), contrasting the migration result when *Fzd7* was silenced (Figure 5A,B). Collectively, these data indicate that Fzd7 receptor play an important role in neuronal migration of the neocortical glutamatergic neurons. Considering that the Fzd7 expressions are shown in VZ region (Figure 2A) and its dendrites (Appendix A), it is possible that the *Fzd7* changes may affect neuronal migration from the VZ region containing neocortical stem cells. 

When we analyzed the expression of Foxp proteins (Figure 6), we surprisingly found that *Fzd7* overexpression resulted in an increase in Foxp2 protein in the upper bins 1–5 (Figure 6A and middle in Figure 6B) while there was no change in Foxp2 expression in the lower bins (right in Figure 6B). Although the Foxp2 protein is normally not expressed in bins 1–3 at P0 neocortex (Figure 2B and Figure 6A), the Fzd7 receptor autonomously induced Foxp2 expression even in neurons of these upper layer bins when *Fzd7* was overexpressed (arrows in Figure 6A), which suggests that the change of Foxp2 expression is not due to the defects in migration. These observations additionally imply that Fzd7 regulates the expression of Foxp2. Furthermore, the *Fzd7* overexpression decreased the Foxp1 expression in GFP^+^ neurons (Figure 6C,D). Interestingly, this reduction in Foxp1^+^/GFP^+^ cells upon the *Fzd7* overexpression was found to be selectively in lower bins 6–7 (arrowhead region in Figure 6C and right in Figure 6D) without the significant change in Foxp1 expression in the upper bins (middle in Figure 6D). Collectively, these findings further support the notion that Fzd7 regulates Foxp2 and Foxp1 expression in developing neocortical neurons.

### 3.6. Wnt3-Fzd7 Signaling Regulates Foxp2 mRNA Translation via Its 3′UTR

The 3′ untranslated region (3′UTR) functions principally on modulating mRNA dissociation with ribosome and translation efficiency [106], including *Foxp2* 3′UTR [48]. We previously reported that the specific 3′UTR sequence of *Foxp2* mRNA regulates translation efficiency in response to Wnt3 signaling in the developing neocortex [48]. First, we repeated these findings in a new set of similar experiments and found that Wnt3 induces mRNA translation via *Foxp2_3′UTR* also in neuronal N2a cells (*Ctrl* shRNA, W+S− in Figure 7). To test whether this Wnt3 induced translation of *Foxp2* mRNA occurs through the Fzd7 receptor, the control (*Ctrl*) or *Fzd7* shRNA plasmid was transfected into the N2a cells together with the *Luciferase*-*Foxp2*_*3′UTR* reporter construct (Figure 7). Twenty-four hours later, either mock (nothing added), recombinant Wnt3, or recombinant Wnt3 + recombinant SFRP1 (WNT inhibitor) was added to the N2a cell culture medium. After 48 h of the treatment, Luciferase protein activities (RLUs) were measured from total protein extracts of each experimental condition and normalized by the *Luciferase* mRNA levels (Figure 7) to examine the translational effect (protein/mRNA ratio) [107]. As shown in the primary neuronal cell cultures [48], the neuronal N2a cells also exhibited that treatment with Wnt3 induces translation via the *Foxp2–3′UTR* (*Ctrl* shRNA, W+S− in Figure 7) and that this translationally enhanced effect is abolished by the SFRP1 inhibitor (*Ctrl* shRNA, W+S+ in Figure 7), representing the Wnt3−specific effect to the *Foxp2* translation.

Importantly, the *Fzd7* silenced cells had reduced *Foxp2* translation (*Fzd7* shRNA, W–S– in Figure 7) compared to the *Ctrl* shRNA-transfected cells (*Ctrl* shRNA, W–S– in Figure 7). In addition, the effect of Wnt3 or inhibitor SFRP1 on *Foxp2-3′UTR* translation did not occur when *Fzd7* was silenced (*Fzd7* shRNA in Figure 7). This reduced effect in Wnt3-induced *Foxp2* translation is consistent with our observations above which show the decrease in Foxp2 protein expression by *Fzd7* shRNA in primary neuronal cells in vitro and developing neocortical neurons in vivo (Figure 4 and Figure 5). Collectively, these data indicate that the Fzd7 receptor functions as a downstream factor for the ligand Wnt3 signaling in timed mRNA translation events of developing neocortices.

## 4. Discussion

Here we summarize the current state of knowledge on the role of extracellular factors for the intrinsic mRNA translation. In addition, we report that Fzd7 receptor is downstream of the extracellular Wnt3 regulation to *Foxp2* mRNA translation. The Fzd7 receptors are expressed broadly throughout the CP layers and colocalized with the Wnt3 ligands (Figure 2) which are secreted from the thalamus at E15 [48]. These Fzd7 proteins also overlapped with distinct Foxp1 and Foxp2 expressions in the upper and lower layers, respectively (Figure 2). In neuroblastoma N2a cells, the Wnt3 intensity at the cell membrane decreases with the *Fzd7* downregulation (Figure 3). Our studies above are in agreement with previously reported Wnt3-Fzd7 binding in intestinal stem cells [103] and the functional interaction between Wnt3 and Fzd7 in hepatocellular carcinoma cells [104]. Collectively, these findings indicate that neocortical Fzd7 receptor can transfer the extrinsic WNT signaling arriving from the thalamus to the neocortex in order to regulate the mRNA translation in developing neocortical neurons, which thus may promote neurogenesis of neocortical cells with regulation of developmental transcription factors Foxp1/2 expressions. The Fzd7 is not previously known to regulate translation and neocortical development.

WNT gradient is one of the most important effectors in deciding the developmental stages [108,109]. Our data represent that Wnt3 secreted from the thalamus works with the neocortical Fzd7 receptor, which then regionally regulates the protein levels of neocortical Foxp transcription factors. Therefore, this Wnt3-Fzd7 axis may contribute to the final steps of neuronal maturation, which include circuit formation and synaptogenesis. Forkhead box (Fox) proteins, Foxp1 and Foxp2, are distinct transcription factors which possess different functions in the specific layers of the neocortex [110,111,112,113]. They are strongly linked to human neurodevelopmental disorders (NDDs), including autism spectrum disorder (ASD), intellectual disability (ID), and speech and language disorder [114,115,116]. While FOXP2 mutations predominantly impair speech and language, FOXP1 mutations cause a severe global NDD [112,113]. Therefore, tight control of their expression levels in developing neurons is critical for neurodevelopment. In primary neocortical neuronal cell cultures, it was found that *Fzd7* downregulation reduces Foxp2 and induces Foxp1 expression, respectively (Figure 4). These changes in Foxp2 and Foxp1 expressions were also found in the developing neocortex when *Fzd7* was downregulated in vivo (Figure 5). In the developing neocortex, this Wnt3-Fzd7-Foxp1/2 signaling can be involved in curing the abnormalities which are associated with NDDs.

The thalamus-specific deletion of *Wnt3* reduces the neocortical expression of Foxp2 in lower layers while the Foxp1 protein expression in the upper layers seems to be unaffected in these mice [48]. In addition, the translational effect for Foxp1 protein is absent in Wnt3 treated cells [48]. However, our data showed that the Fzd7 receptor affects both Foxp2 and Foxp1 expression in primary cells and neocortices (Figure 4, Figure 5 and Figure 6), suggesting that the Fzd7 receptor differentially regulates Foxp1 and Foxp2 expression, and that a different ligand possibly acts on the upper layers of Foxp1 than in lower layers of Foxp2. In line with the previous report [48], here we found that Foxp1 expression remains unchanged in primary neuronal cell cultures exposed to Wnt3 ligand (Appendix A). Nonetheless, the *Fzd7* shRNA effect resulting in Foxp1 increase still occurred in both Wnt3 treated and Wnt3 non-treated cells (Appendix A). These findings suggest that Foxp1 expression is regulated by the signaling of the Fzd7 receptor, but not by the Wnt3 ligand. A recent report showed that the Fzd7 receptor can physically and functionally interact with the Wnt7b ligand in the hippocampus to modulate dendritic growth and complexity [117]. We observed strong signals of Fzd7 in the dendrites of neocortex when *Fzd7* was overexpressed (Appendix A). It is possible that the Fzd7 receptor in the neocortex may interact with several Wnt ligands to integrate the diverse signals for the complexity in neocortical development.

This study as well as previous findings emphasize the importance of timed extracellular signals in neuronal development, including the neocortex. The exact control of protein expression levels at distinct developmental time points serves the needs of developing cells. The interplay of extrinsic signals and intrinsic protein synthesis rates is critical for normal neurodevelopment. It is easy to envision that slight alterations in these signaling cascades may result in devastating abnormalities within the central nervous system, which would manifest as NDDs.

## Figures and Tables

**Figure 1 cells-10-00253-f001:**
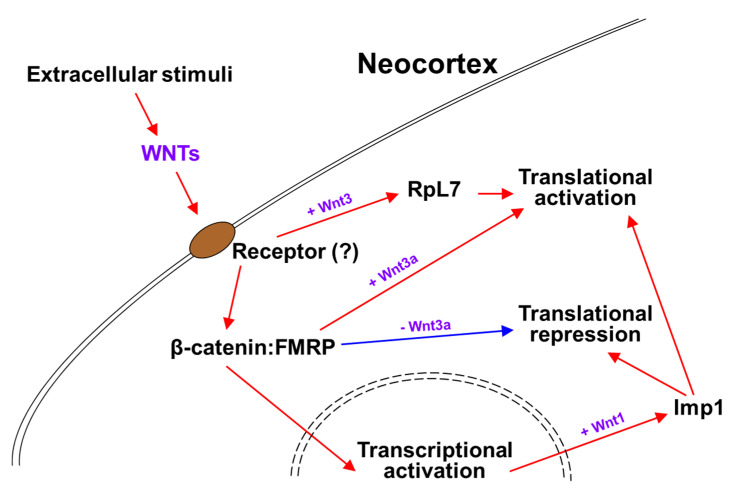
mRNA translations regulated by WNTs signaling. (1) Wnt3, secreted from thalamocortical axons, increases Rpl7 ribosomal protein within polysomes and promotes Foxp2 mRNA translation in the neocortex [48]. (2) Wnt3a stimulation induces the translational activation of mRNAs by sequestering β-catenin away (red arrow) from the β-catenin and FMRP complex that sustains the translational repression state (blue arrow) [97]. (3) Wnt1 signaling enhances RBP Imp1 expression during cortical development, which both translationally silences proneurogenic mRNAs and increases expression of pro-self-renewal mRNAs, to maintain NSCs in an undifferentiated state [98].

**Figure 2 cells-10-00253-f002:**
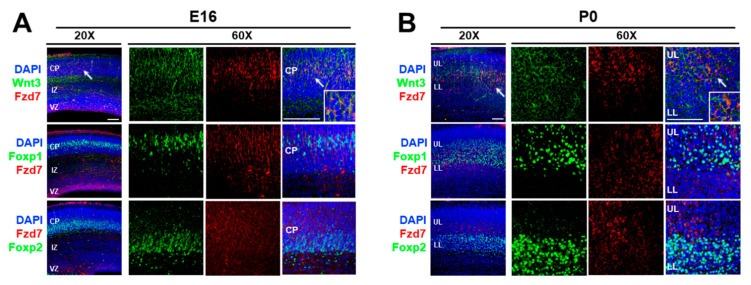
Colocalization of Wnt3 and Fzd7 with Foxp1 and Foxp2 expression at E16 and P0. (**A**,**B**). Immunohistochemistry (IHC) for Wnt3 (green), Fzd7 (red), Foxp1 (green), and Foxp2 (green) in WT E16 (**A**) and P0 (**B**) neocortices (*n* = 5 animals per age). Arrow denotes colocalized area between Wnt3 and Fzd7 in confocal microscope image using 20× objective lens, which was confirmed with 60× resolution (insert: higher magnification image of colocalization). DAPI shown in blue; CP: cortical plate; IZ: intermediate zone; VZ: ventricular zone; UL: upper layers; LL: lower layers; scale bar: 100 μm.

**Figure 3 cells-10-00253-f003:**
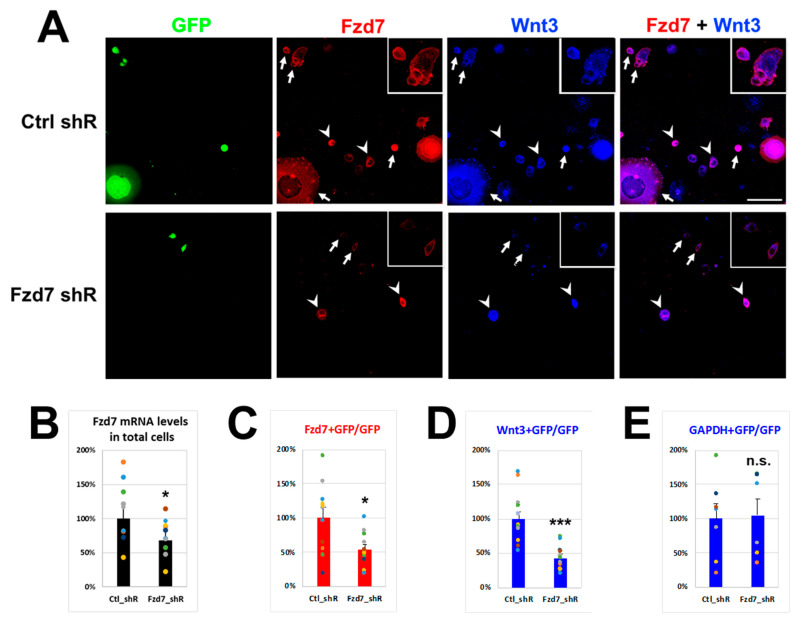
Interaction between Wnt3 and Fzd7 in neuroblastoma N2a cells. (**A**). IHC for GFP (green), Fzd7 (red), and Wnt3 (blue) in N2a cells after 3 days of each shRNA transfection containing *GFP* marker (*Ctrl* or *Fzd7* shRNA) and following 4 h of Wnt3 (100 ng/mL) treatment (*n* = 3 separate transfections). Arrow: shRNA-transfected GFP cells (insert: higher magnification image); arrowhead: non-transfected cells; scale bar: 100 μm. (**B**). *Fzd7* mRNA expression levels in total N2a cells determined by qRT-PCR (*n* = 10 repeats) when IHC was performed in **A**. Data of *Fzd7* shRNA were normalized to Gapdh and then to *Ctrl* shRNA (100%). (**C**–**E**). Quantification of Fzd7 (**C**), Wnt3 (**D**), Gapdh (**E**)-positive cells among total GFP cells (251–356 counted), normalized to the *Ctrl* shRNA (*n* = 6–12 repeats). Data represent the mean and SEM. Statistics: Student’s *t*-test; *: *p* < 0.05; ***: *p* < 0.005; n.s.: not significant.

**Figure 4 cells-10-00253-f004:**
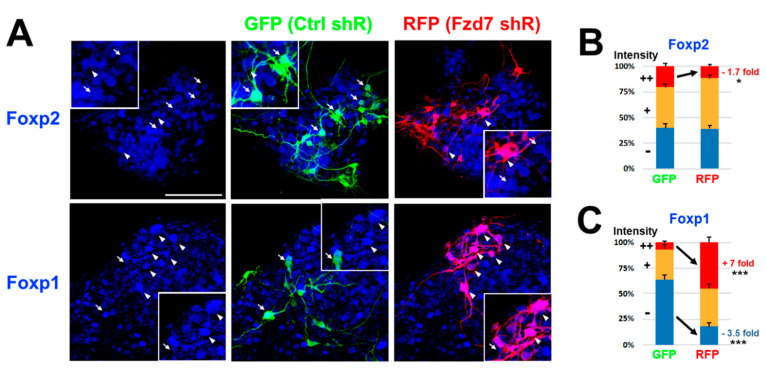
Foxp2 downregulation and Foxp1 upregulation with *Fzd7* shRNA in primary neuronal cells. (**A**). IHC for GFP (*Ctrl* shRNA cells: green) and RFP (*Fzd7* shRNA cells: red) with Foxp2 or Foxp1 immunostaining (blue) after four days in vitro (DIV) culture of primary neuronal cells in utero electroporated (IUE) at E13 (*n* = 3 animals of each GFP and RFP). GFP (arrow) and RFP (arrowhead) cells showing strong (++) intensity of Foxp2 (lower panel: Foxp1) protein are respectively represented. Insert: higher magnification images of same area; scale bar: 100 μm. (**B**,**C**). Quantification of Foxp2 (**B**) or Foxp1 (**C**) signal with −, +, and ++ intensities which were compared/normalized to the intensity (+) observed in non-transfected cell as an intrinsic control of immunostaining quality (*n* = 28–48 pictures counted with total 403–700 cells). Data (%) represent the mean and SEM of each intensity (total sum = 100%). Statistics: Student’s *t*-test; *: *p* < 0.05; ***: *p* < 0.005.

**Figure 5 cells-10-00253-f005:**
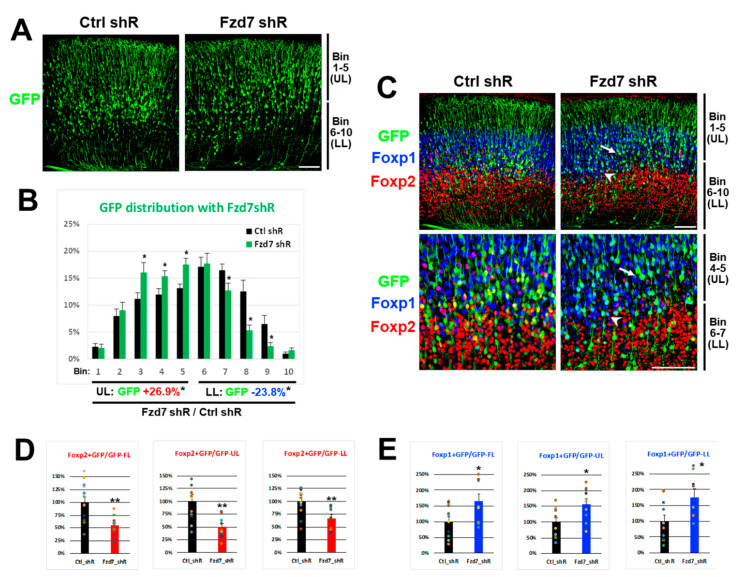
Foxp2 downregulation in UL and Foxp1 upregulation in LL with *Fzd7* shRNA at P0 (E13–P0). (**A**)**.** IHC for GFP cells (green) in P0 neocortices electroporated in utero with *Ctrl* or *Fzd7* shRNA at E13 (*n* = 3 animals). Bin 1–5: upper layers (UL); Bin 6–10: lower layers (LL); scale bar: 100 μm. (**B**). Percentage (%) of GFP^+^ cells in each bin from total 10 bins (*n* = 13–15 pictures counted with total 1965–2500 GFP cells). GFP^+^ cells in the upper layers (Bin 1–5) of *Fzd7* shRNA neocortices increased up to 26.9% compared to *Ctrl* shRNA, while decrease of 23.8% was shown in the lower layers (Bin 6–10). *: *p* < 0.05. (**C**). IHC for GFP (green), Foxp1 (blue), and Foxp2 (red) in P0 neocortices. Arrow and arrowhead: the areas showing bigger change in Foxp2 and Foxp1 expression, respectively, which are shown with higher magnification images in the lower panel. (**D**,**E**). Quantification of Foxp2 (**D**) and Foxp1 (**E**)-positive cells (*n* = 8–14 pictures counted with total 1752–2358 GFP cells), normalized to the *Ctrl* shRNA (100%). Data represent the mean and SEM. FL; full layers (Bin 1–10); statistics: Student’s *t*-test; *: *p* < 0.05; **: *p* < 0.01.

**Figure 6 cells-10-00253-f006:**
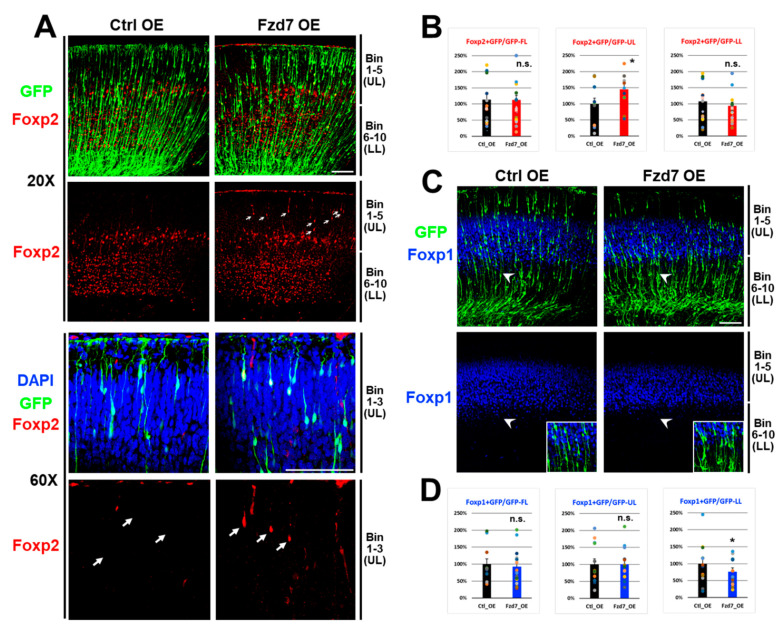
Foxp2 upregulation in UL and Foxp1 downregulation in LL with *Fzd7* OE at P0 (E16–P0). (**A**,**C**). IHC for GFP cells (green) with Foxp2 (red in **A**) or Foxp1 (blue in **C**) immunostaining in P0 neocortices electroporated in utero with *Ctrl* or *Fzd7* OE plasmid at E16 (*n* = 3 animals). Arrow in **A**: Foxp2 protein expressed in upper layer bins 1–3 with the *Fzd7* overexpression; arrowhead area in **C**: Foxp1 expression reduced in lower layer bins 6–7 with the *Fzd7* overexpression, which are shown with higher magnification images in the insert. DAPI shown in blue (**A**); Bin 1–5: upper layers (UL); Bin 6–10: lower layers (LL); scale bar: 100 μm. (**B**,**D**). Quantification of Foxp2- (**B**) or Foxp1- (**D**) positive cells (*n* = 10–20 pictures counted with total 1711–3358 GFP cells), normalized to the *Ctrl* shRNA (100%). Data represent the mean and SEM. FL; full layers (Bin 1–10); statistics: Student’s *t*-test; *: *p* < 0.05; n.s.: not significant.

**Figure 7 cells-10-00253-f007:**
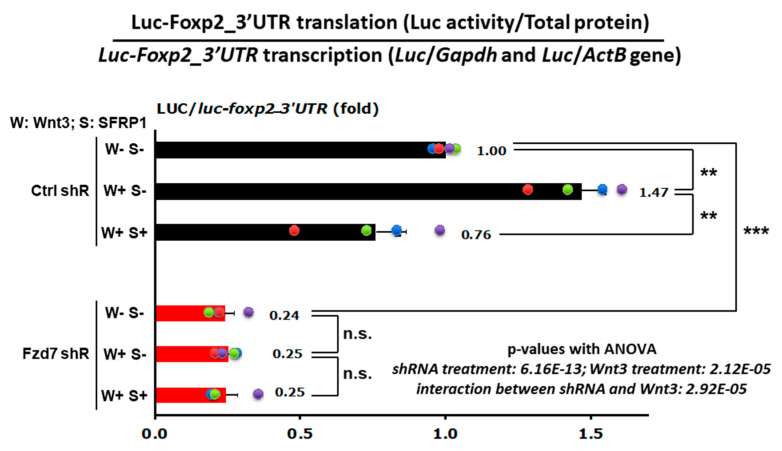
Translational reduction through *Foxp2–3′UTR* with *Fzd7* shRNA. Quantification of translational effects (protein/mRNA ratio) was carried out using *Luciferase*-*Foxp2*_*3′UTR* construct in neuronal N2a cells, transfected with the *Ctrl* or *Fzd7* shRNA plasmid and treated with mock (W–S–), Wnt3 (100 ng/mL) (W+S–), or Wnt3 + SFRP1 (100 ng/mL) (W+S+) for 48 h (*n* = 4 repeats). Translation was measured with Relative Luciferase light units (RLUs), and was then normalized to the *Luciferase* mRNA level (qRT-PCR). Data represent the mean and SEM. Statistics: Student’s *t*-test; **: *p* < 0.01; ***: *p* < 0.005; n.s.: not significant. Two-way ANOVA analysis with replication represents that the *Fzd7* shRNA and Wnt3 treatments are significant and that the effects of them are dependent on each factor.

## Data Availability

Not applicable.

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
