# Peer review of "Extrinsic Regulators of mRNA Translation in Developing Brain: Story of WNTs"

_cells, 2021, doi:10.3390/cells10020253_

Round 1
Reviewer 1 Report
In this study, Park et al. analyzed the importance of Fzd7 as a mechanism to regulate the expression of Foxp1 and Foxp2 in the neocortex. In their previous report, they reported that Wnt3, secreted from TCA, plays an important role in the translation of Foxp2 in the neocortex. In this paper, they found that Fzd7 bound to Wnt3 in N2a cells, that knockdown of Fzd7 reduced the number of Foxp2-positive cells, and that overexpression of Fzd7 increased the number of Foxp2-positive cells. Interestingly, we also found that Fzd7 decreased the expression of Foxp1. In addition, reporter assay using N2a cells showed that Fzd7 suppressed the 3'UTR of Foxp2. In conclusion, we argue that Fzd7 induces Foxp2 expression as a receptor for Wnt3
The effects of Fzd7 on Foxp1 and Foxp2 positive cells are well analyzed and the importance of Fzd7 is well understood. However, there are insufficient experiments on two points: whether Fzd7 interacts with Wnt3 and whether Fzd7 regulates Foxp2 translation. It is also unclear whether knockdown or overexpression of Fzd7 regulates Foxp1 and Foxp2 expression itself, or whether Foxp1 and Foxp2 expression is altered via effects on migration or changes in neural subtypes. Therefore, several important experiments need to be performed before publication.
Major points
- It is difficult to understand how important it is in the field of neurogenesis to show that Fzd7 is a receptor for Wnt3 in the mechanism of regulation of Foxp2 expression. They should discuss more its importance.
- Although they claim that Fzd7 regulates the translation of Foxp2, in Fig7 the 3'UTR can regulate not only translation but also mRNA degradation and localization. In addition, in Fig. 5 and Fig. 6, we can see that Fzd7 controls the translation of Foxp2. They should either correctly examine whether Fzd7 regulates the translation of Foxp2 or rewrite the text to include the title. They should also show that there is no change in the amount of Foxp2 mRNA in Fig. 5 and Fig. 6.
- The experiments in Fig. 3 and Fig. 7 using N2a should also be performed in vivo.
- In Fig. 5, they should examine in more detail whether the change in Foxp1 and Foxp2 expression is due to the defects in migration or change in the neuronal subtype. For migration, they need to analyze at an earlier stage. For the neuronal subtype, staining with layer markers different from those of Foxp1 and Foxp2, such as Tbr1, Ctip2, Rorb, Cux1, and Brn2, should be performed.
Minor points
- In the abstract, there is no evidence that Fzd7 regulates Foxp1 “translation”, so it should be rewritten.
- The presentation of the fluorescence images should be improved. In Fig. 2 and Fig. 3, it is unclear whether they are co-localized or not. Therefore, higher magnification images and quantification of colocalization are needed. In Fig. 4, Fig. 5, Fig. 6, and Fig. S2, the area indicated by the arrow are not clear. Therefore, higher magnification images should be shown.
- In Fig. 3, an experiment without Wnt3 should also be performed as a negative control for Wnt3 staining.
- In Fig. 5, Fzd7 knockdown should be confirmed in vivo since the cell type is different from N2a in Fig3.
- It is interesting to examine whether the Wnt7b treatment decreases Foxp1 expression.
- In line 553, “mRNA”. In line 610, “downregulated”.
Reviewer 2 Report
This manuscript by Park et al is focused on Wnt regulation of mRNA translation of Foxp transcription factors during neocortical development, specifically focused on the Fzd receptor expressed by nascent projection neurons mediates this effect. The authors use a very good combination of cell culture and in vivo electroporation to convincingly demonstrate that Fzd7 is the main mediator of Wnt3’s effect on Foxp expression during corticogenesis. Below are moderate concerns regarding description of methods, statistical analysis and presentation of data in the figures that need to be addressed. Also, the extended review in the introduction (see point #1 below) is a weakness of the manuscript, better to focus on information necessary to understand the relevance of the data presented in this manuscript, save the review material for another paper.
Major:
- The authors have combined a primary research article with a review, this is presented as an extended introduction covering neocortical development and Wnt signaling in neuronal diseases and regulation of mRNA translation. From a style perspective, it would make the manuscript more readable if it had much shorter introduction just presenting background relevant to the data presenting. There is quite a bit of extra information here that detracts from the main point of the manuscript, Wnt-Fzd regulated expression of Fox TFs in neocortical development. The remaining material could be made into a separate review document.
- Fig 3: 1) An additional control for this experiment should antibody staining for Wnt3 with no Wnt3 treatment. This would confirm the antibody staining is detecting exogenous added Wnt3. 2) The Fzd5 shR image in 2B has very few cells, please include an image that is representative of an image quantified (ex: similar to what it shown in supplemental fig 1B). 3) The scale bar is listed as 100 microns however some of the cells appear to be over 100 microns in size (which seems off), please check accuracy of listed scale bar length in figure legend.
- Fig 4: 1) It is not clear from the methods or results how a value of ‘-‘, “+” or “++” was defined, does this reflect a set range of Foxp1 or Foxp2 intensity for a given RFP+ or GFP+ cell? Please provide additional information in the methods section as to how quantification was performed. 2) Higher magnification insets are required for the reader to see individual cells. With these images, there is not an obvious difference in intensity in Foxp1 or Foxp2 between RFP, GFP and non-transfected cells. 3) Two n are reported in the figure legend but it is not clear which was used for statistical tests. For example, n=3 is the number of biological replicates (possibly refers to 3 separate electroporation experiments where brains were pooled from a complete litter?) but also reporting n as the number of images used for analysis. The n=3 is the correct as this is the biological replicate. This is different than the technical replicate, multiple pictures from a single animal. Please confirm that this is the case (n=3) or redo the statistical analysis to reflect this). Minor: there is box legend on Fig 4C, I don’t think that is meant to be there? Please correct.
- Figure 5, 6: 1) Similar concern as to Fig 4 with regard to replicates, two different ‘n’ are reported in figure legend, n=3 is correct but the graphs indicate that the replicate used is counts per image. Stats need to be redone to reflect biological replicate (n=3). 2) 5C and 6C, it is not clear what the arrows in the images are pointing to, high magnification insets are needed (like something similar to 6A, Foxp2/GFP cells in the UL)
- There is not any mention in the discussion of the migration phenotype show in the Fzd OE. Are the cells mis-migrating because of altered Foxp expression or is this a different function of Wnt-Fzd? Connecting this back to the phenotypes (mis-migration in Wnt3 KO from thalamus in their previous work?) would be helpful.
Minor:
Ln 11-12: typo, I think it should read “..to be the Wnt3 receptor downstream in Wnt3-regulated mRNA translation.”
Ln 25-26: Per comment #1 in ‘major’, edit last sentence of abstract.
Ln553: remove extra space between “m” and ‘RNA’
Ln584: “is downstream” (delete ‘a’)
Reviewer 3 Report
Park et al., reports a potential mechanism underling Wnt3 mediated Foxp translation in the neurons in the cerebral cortex. The authors did multimodal evaluation and most of data look solid. A major concern is that the authors might submit the draft accidentally. Extensive English editing is required. Some of them are found below.
- Supplementary Figures; no legends. Therefore it is impossible to judge the accuracy of content.
- Introduction; 4 subdivisions. Can make it concise.
- Fig1: revise/remove "neocortex" as it is inside cell. and remove extracellular stimuli.
- Fig2: add s after len in legend. Hard to see colocalization of signals in images.
- Fig3:B "fzd7 mRNA in total cells" probably means "fzd7 mRNA positive cells in total cells".
- Some places say shR, others say shRNA. shR was not defined.
- Fig4: C includes weird Chinese character and so on, that need to be removed.
- Fig5 legend: "higher change" needs to be changed to "bigger change"
- Fig6: remove at P0 (E16-P0) in title. C: single blue color can be changed to white color, so that the differences are easily found.
- Fig7: I am afraid t-test is inappropriate. The interaction between treatment and gene manipulation needs to be tested.
Round 2
Reviewer 1 Report
Many of the concerns have been improved through the great efforts of the authors. On the other hand, the following points need to be further improved.
Related to Major #1 comment, they should describe how the discovery of Fzd7 as a receptor for Wnt3 is important in this field before or after "However, the WNT receptor that may mediate these mRNA translation events remains unknown." in the Introduction. In the current version, the readers have to read the Results without recognizing the importance of the experiment.
Related to Major #3 comment, they should describe the in vivo confirmation experiment as a future study since the experiment can be performed by isolation of GFP-positive cells using FACS.
Related to Major #4 comment, there is still a possibility that Fzd7 knockdown, not Fzd7 overexpression, causes delayed neuronal migration and changed the type of neurons, from Foxp2-positive to Foxp1-positive. They should include this point in the Discussion.
Author Response
Please, see the attachment.

Reviewer 2 Report
There continue to be a few problems that need to be addressed by the authors, the stats were not adequately addressed and a few modifications were not performed as requested.
1) The authors have opted to not refine the introduction. I do not agree with this. I believe the extended introduction impacts readability of the work and detracts from the manuscript. However I will leave this up to the editor as it is a stylistic critique and does not relate to the rigor or accuracy of the primary data.
2) Figure 4: The authors only provided higher magnification images of the Foxp1 and Foxp2, without the GFP or RFP with Foxp1 and Foxp2 it is hard to tell based on arrows alone which cells have the Fzd7 shRNA (RFP) vs Control (GFP) snRNA. I would like to see insets for all panels.
3)The author’s removed the n value referring to the images however the graphs in Fig. 4, 5, 6 still show multiple data points suggesting that they are using individual images for statistical analysis. This is not correct. Statistical analysis should reflect biological replicates, so there should be 3 data points, one for each embryo. Each data point should be an average of counts taken from images in a single experimental embryo (n=3). And the statistical tests need to be redone with correct values and replicates (n=3) and new p values reported.
4) Fig 5 and 6: Putting the high mag insets in a separate figure in the supplemental is not helpful to the reader, these need to be in the main figure, specifically Supp. Fig 2C should be next to Fig. 5C. And a high magnification inset for Fig. 6C was not provided.
Author Response
Please, see the attachment.

Reviewer 3 Report
Manuscript has significant improvement from previous version.
Author Response
Please, see the attachment.
